# Optimization of the Wire Diameter Based on the Analytical Model of the Mean Magnetic Field for a Magnetically Driven Actuator

Zhangbin Wu , Hongbai Bai, Guangming Xue * and Zhiying Ren

School of Mechanical Engineering and Automation, Fuzhou University, Fuzhou 350116, China
* Correspondence: yy0youxia@163.com; Tel.: +86-150-0514-1625

**Abstract:** A magnetic field induced by an electromagnetic coil is the key variable that determines the performance of a magnetically driven actuator. The applicability of the empirical models of the coil turns, static resistance, and inductance were discussed. Then, the model of the mean magnetic field induced by the coil was established analytically. Based on the proposed model, the sinusoidal response and square-wave response were calculated with the wire diameter as the decision variable. The amplitude and phase lag of the sinusoidal response, the time-domain response, steady-state value, and the response time of the square-wave response were discussed under different wire diameters. From the experimental and computational results, the model was verified as the relative errors were acceptably low in computing various responses and characteristic variables. Additionally, the optimization on the wire diameter was carried out for the optimal amplitude and response time. The proposed model will be helpful for the analytical analysis of the mean magnetic field, and the optimization result of the wire diameter under limited space can be employed to improve the performance of a magnetically driven actuator.

**Keywords:** mean magnetic field; wire diameter; coil; sinusoidal response; square-wave response





## 1. Introduction

The magnetically driven actuator has been widely used in plenty of engineering fields, including vibration reduction or control, ultra-precision machining, acting fluidic valves, etc. [1–4]. Magnetically driven actuators have also been introduced to quite commonly actuate an aerospace device [5–15], including the electro-hydraulic servo valve.

Optimization of the actuator is quite important to improve the actuator's performance. A magnetic field was generally chosen as the optimization objective function as it influences the output performance of the actuator directly, and is the simplest variable to optimize the actuator, compared to the magnetization/magnetic induction intensity or the displacement. Taking the giant magnetostrictive actuator which employs the giant magnetostrictive material (GMM) as its actuation core as an example, Figure 1 summarizes the generally used optimization methods. From the point of view of magnetic fields, the optimization of actuator performance was generally converted to the promotion of the mean magnetic field in the GMM area, which is equivalent to the maximization of the magneto motive force (MMF) distributed on GMM. Additionally, two methods were used to promote the MMF on GMM, respectively, improving the MMF ratio occupied by GMM and increasing the total MMF.

The first optimization method was accomplished based on some magnetic field models from a "field" or "circuit" method [16–19]. Liang Yan et al. [20] and HyoYoung Kim et al. [21] proposed a mathematic model based on the Biot–Savart law and the finite element model to formulate the three-dimensional magnetic field distribution in a spherical actuator. Abdul Ghani Olabi et al. [22] also established the finite element model of a magnetostrictive actuator for analyzing the magnetic field in the actuator. The proposed

models supplied the mean values and distribution characters of magnetic devices, which were quite helpful for the magnetic circuit optimization. Due to the complex magnetic circuit of the hybrid excitation generator used in an energy conversion system, Huihui Geng et al. [23] proposed an analytical method of the main magnetic field, where the Carter coefficient and rotor magnetomotive force were taken as the objective variables. Compared with traditional methods, the proposed method can improve the accuracy of the outputted magnetic field. Jaewook Lee et al. [24] adopted a simplified finite element model to execute structural topology optimization for the high magnetic force of a linear actuator, and they found that the use of a periodic ladder structure was best for magnetic field manipulation. Kim Tien, Xulei Yang et al. [14,25] utilized the finite element model to analyze the distribution of the magnetic field in a giant magnetostrictive actuator separately. By adjusting the permeability of the parts appropriately, the uniformity and mean intensity of the magnetic field within the material could be improved. Some other modeling and optimizing methods for the magnetic field within specified structures can also supply effective references [17,26–29]. On the whole, the circuit model was always used to form a magnetic field model for an analytical analysis. The finite element model [19,30,31] was commonly used to promote magnetic field uniformity. For the mean magnetic field applied to the giant magnetostrictive material, the positively proportional model vs. the coil current [1–3,16,19,28,32–35] was quite commonly used. Then, the closed circuit was verified to be helpful for higher magnetic field intensity [15] as it improved the proportional factor.

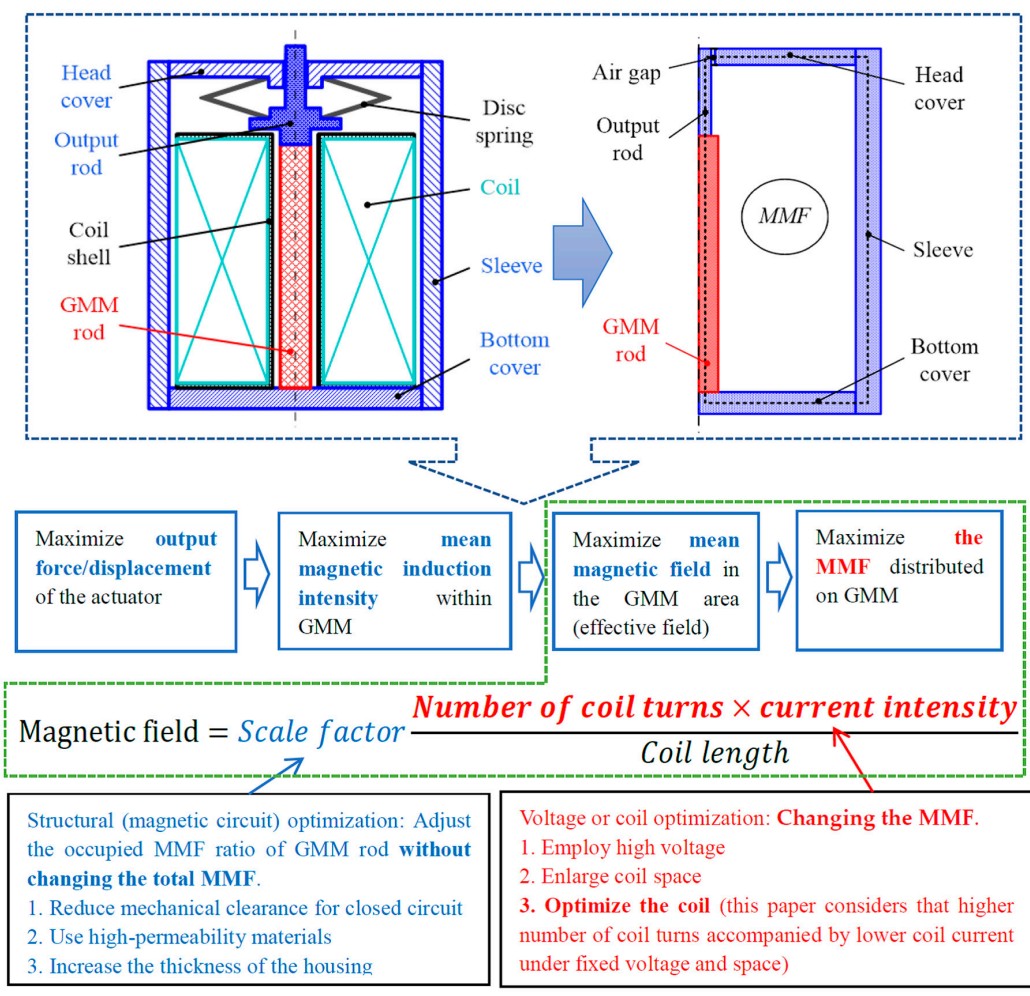

**Figure 1.** Generally used optimization methods for the giant magnetostrictive actuator: Refs. [16–31] based on the first method and Refs. [36–44] based on the second one.

For the second optimization method, an appropriate voltage waveform or the winging method was considered to directly promote the total MMF and the magnetic field induced by a coil [2]. A high threshold with a low-holding voltage has been widely used in an electromagnetic injector. C.B. Britht et al. [36] and G. Xue et al. [37,38] introduced this type of voltage to stimulate a giant magnetostrictive device. Additionally, it was comprehensively verified that the introduced voltage promoted the response time of the coil current, and the magnetic field, quite efficiently. Manh Cuong Hoang et al. [39] proposed an optimization method of the magnetic field for an electromagnetic actuation system. The maximum magnetic and gradient fields were significantly enhanced by the proposed algorithm compared to the conventional independent control. Haoying Pang et al. [40] proposed a novel spherical coil for the atomic sensor, where the magnetic field uniformity was improved along the axis. Yiwei Lu et al. [18] introduced the magnetically shielded room to enhance the coil magnetic field and reduce power loss for a multi-coil system. Cooperated with the non-dominated sorting genetic algorithm, the design reached prominent reductions in total current and power loss. Yundong Tang et al. [41] introduced two correcting coils to improve the uniformity of the magnetic field for a solenoid coil, while it was not so convenient when the coil space was limited as the correcting coils should have occupied some axial spaces. Some other optimization methods for the coil or contactor [42–44] can also provide useful references for optimizing the magnetic field induced by an electromagnetic coil.

Based on the second optimization method, this paper focuses on coil optimization when the volume of the magnetically driven actuator suitable for an electro-hydraulic servo valve is limited. In this paper, the dynamic magnetic field was tested based on the linear relationship between the magnetic field and coil current. Then, the dimension parameter, static resistance, and static induction were modeled based on empirical equations or mathematical fitting. The mean magnetic field within the coil was modeled, especially its functional relationship with respect to the wire diameter. Then, the sinusoidal and square-wave responses were calculated, and the important characteristic parameters of these responses were extracted. From the calculated and tested results, the influence of the wire diameter on the mean magnetic field was discussed comprehensively for an optimal selection of the wire diameter. During analysis, the relative errors in computing various variables were also given to verify the precision of the proposed model and effectiveness of the optimization. For the magnetically driven actuator, optimized results can be employed to promote the amplitude and response speed of the mean magnetic field, and then to improve the actuator performance.

## 2. Experimental Methods

### 2.1. Test Principle

The dynamic magnetic field intensity or magnetic flux density was always measured "indirectly" based on Ampere's circuital theorem or Faraday's law of induction. Based on the former principle, the induced magnetic field and coil current have an ideal positive proportion relationship, which has been a commonly used analytical model of the mean magnetic field in some magnetically driven actuators, especially the giant magnetostrictive actuator. Based on this measuring principle, as long as the coil current is measured, the accurate mean of the magnetic field in a dynamic type can be obtained.

The model was easily given by adding a proportional coefficient to the magnetic field model in an infinitely long solenoid [3,5,14,16,19,33–35]

$$H = C_{HI}\frac{NI}{L} \tag{1}$$

where $H$ is the magnetic field intensity and $I$ is the current intensity within the coil, $C_{HI}$ is the proportional coefficient of the mean magnetic field intensity; its value belongs to (0,1), $N$ is the number of the coil turn, and $L$ is the coil length.

The following optimization was based on Equation (1)—the optimization is effective as long as the mean magnetic field in the magnetically driven actuator is in direct proportion

to the product of the number of coil turns and current intensity. For a hollow coil, Equation (1) was not only capable of computing the mean magnetic field within homogeneous medium, but was also suitable to the local mean magnetic field as long as the whole magnetic circuit was filled locally uniformly and did not have too many reluctance numbers. Considering Equation (1) is suitable for most giant magnetostrictive actuators and some micro-displacement electromagnetic actuators; the optimization proposed in this paper is suitable to these types of actuators.

### 2.2. Experiment Setup and Parameters

The experimental system was shown in Figure 2. As illustrated in Figure 2a,c, the computer controlled PS3403D digital oscilloscope (with an embedded signal generator) to generate the required waveform signals. The generated signals were then amplified by an ATA304 power amplifier and inputted into the two ends of the coil. The input voltage at both ends of the coil was differentially collected and the coil current was measured by a TA189A current clamp. The measured voltage and current data were delivered into the digital oscilloscope and then into the computer for processing.

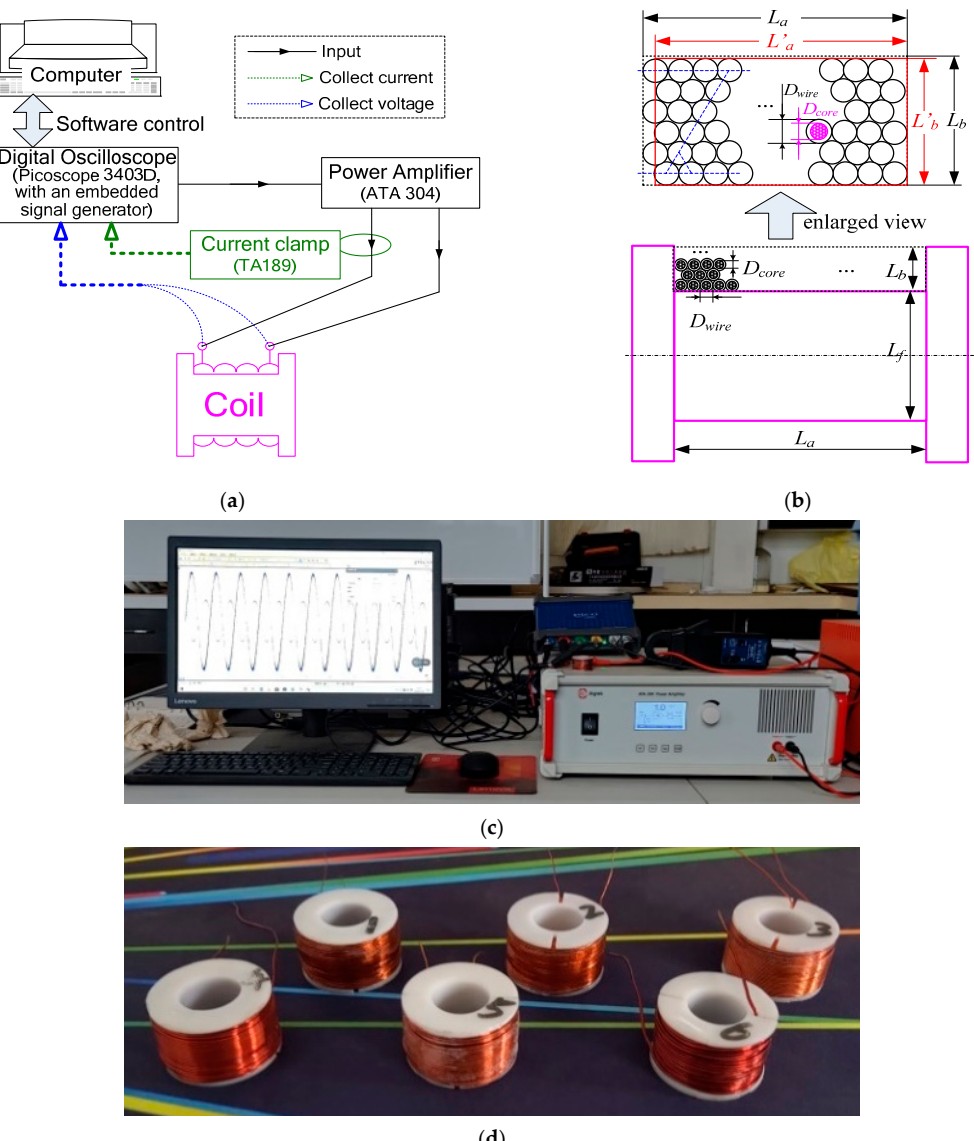

**(a)**

**(b)**

**(c)**

**(d)**

**Figure 2.** Experimental system and wound coils: (**a**) block diagram of the experimental system; (**b**) dimensioned sectional drawing of the coil, (**c**) photograph of the experimental system; (**d**) photograph of the coils.

Figure 2b,d supplied the sectional drawing and photograph of the coils, where $L_a$, $L_b$, $L_f$ represented the coil length, coil thickness, and diameter of the skeleton shaft, respectively, and $D_{wire}$ and $D_{core}$ were the enameled wire diameter and copper core diameter, respectively. The coils were tightly wound by the use of standard enameled wires. Since this article focuses on the optimization of the coil itself, it is not necessary to consider the influence of the iron core or other parts in an actuator. The parameters of the coils are given in Table 1, and some necessary parameters of the skeleton and material are supplied in Table 2. Considering the value of $C_{HI}$ does not affect the increasing or decreasing relationship between the variables; $C_{HI}$ will have no effect on the optimization results. $C_{HI}$ is specified as 0.8 here.

**Table 1.** The main parameters of the coils.

| Coil Label | External Diameter ($D_{wire}$) [mm] | Core Diameter ($D_{wire}$) [mm] | Number of Coil Turns ($N$) [Null] | Resistance ($R$) [Ω] | Inductance ($L$) [mH] |
|---|---|---|---|---|---|
| Coil 1 | 0.31 | 0.27 | 837 | 18.325 | 10.933 |
| Coil 2 | 0.39 | 0.35 | 537 | 7.472 | 4.487 |
| Coil 3 | 0.49 | 0.44 | 342 | 2.994 | 1.789 |
| Coil 4 | 0.60 | 0.55 | 229 | 1.342 | 0.801 |
| Coil 5 | 0.69 | 0.64 | 175 | 0.767 | 0.459 |
| Coil 6 | 0.80 | 0.74 | 124 | 0.410 | 0.243 |

**Table 2.** The main parameters of the skeleton and material.

| Parameter (Variable) [Unit] | Value |
|---|---|
| Coil length ($L_a$) [mm] | 16.5 |
| Coil thickness ($L_b$) [mm] | 6.8 |
| Diameter of skeleton shaft ($L_f$) [mm] | 18.2 |
| Resistivity of copper ($\rho$) [Ω·m] | $1.71 \times 10^{-8}$ |
| Proportional coefficient ($C_{HI}$) [null] | 0.8 |

## 3. Data Processing and Analysis

### 3.1. Inherent Characteristic Parameters of Coils

#### 3.1.1. Dimension Parameters

Standard enameled wire has a nominal diameter of the external wire or the copper core. Then, a certain functional relationship can be supplied between the enameled wire diameter $D_{wire}$ and the copper core diameter $D_{core}$. Figure 3 shows the actual values of $D_{wire}$ and $D_{core}$ and the fitted results using linear functions. It can be seen from Figure 3 that the diameter of copper core is approximately linear vs. the external diameter of enameled wire. With and without an intercept, the fitted linear equations were determined as $D_{core} = 0.9687$ $D_{wire} - 0.03214$ and $D_{core} = 0.9394\ D_{wire}$, respectively. The linear function with an intercept was quite accurate as the relative error was lower than 1.52% when $D_{wire}$ was higher than 0.3 mm and lower than 2.55 mm. In contrast, the linear function without an intercept was not so accurate since the relative error was higher than 5% under some conditions, especially when $D_{wire}$ was quite low.

Though the positively proportional relationship was not suitable to a wide range of dimensions, it may be feasible when the $D_{wire}$ changed within a relatively narrow interval. The coils used in this paper were wound by the wires with diameters of 0.3~0.8 mm. Executing a simple linear fitting, Table 3 supplies the results and relative errors of the two line equations. From computation, the linear equation with intercept was $D_{core} = 0.962\ D_{wire} - 0.0277$ and had a relative error lower than 0.84%. In comparison, the linear equation without an intercept $D_{core} = 0.898\ D_{wire}$ also had high precision as the relative error was lower than 3.2%. Thus, it is acceptable to use a positively proportional function to describe the relationship between $D_{core}$ and $D_{wire}$ when $D_{wire}$ changes within a narrow interval, which is quite convenient for the following optimization.

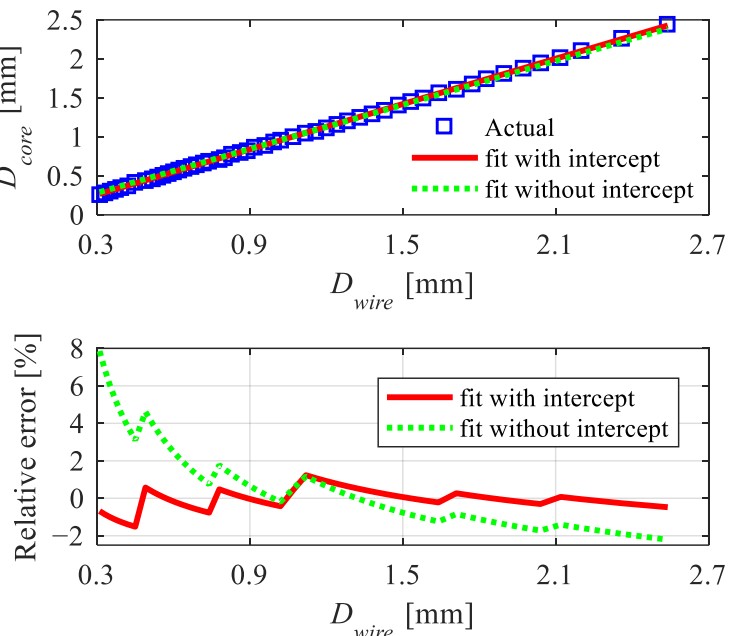

**Figure 3.** Actual and fitted values of $D_{core}$ simultaneously supplied the relative errors of the fitting lines with and without an intercept.

**Table 3.** Linear fitting between $D_{core}$ and $D_{wire}$ when $D_{wire} \in [0.3, 0.8]$.

| $D_{wire}$ | Tested $D_{core}$ | $D_{core}$ from $0.962\,D_{wire} - 0.0277$ | Relative Error of $0.962\,D_{wire} - 0.0277$ (%) | $D_{core}$ from $0.898\,D_{wire}$ | Relative Error of $0.898\,D_{wire}$ (%) |
|---|---|---|---|---|---|
| 0.31 | 0.27 | 0.2705 | 0.1926 | 0.2784 | 3.1037 |
| 0.39 | 0.35 | 0.3475 | −0.7200 | 0.3502 | 0.0629 |
| 0.49 | 0.44 | 0.4437 | 0.8364 | 0.4400 | 0.0045 |
| 0.6 | 0.55 | 0.5495 | −0.0909 | 0.5388 | −2.0364 |
| 0.69 | 0.64 | 0.6361 | −0.6125 | 0.6196 | −3.1844 |
| 0.8 | 0.74 | 0.7419 | 0.2568 | 0.7184 | −2.9189 |

From the sectional drawing shown in Figure 2b, it can be observed that winding a coil was equivalent to arranging the cross-sectional area of the wire in the rectangular area supplied by the coil skeleton. The coil turns must be an integer; while $L_a$ or $L_b$ was not exactly the integral multiple of $D_{wire}$, the effective length $L_a{'}$ and thickness $L_b{'}$ were a little lower than $L_a$ and $L_b$, respectively. Coil length or thickness was not fully utilized, and the available area was $L_a{'} \times L_b{'}$, which was slightly less than the actual area.

From Figure 2b, the turn number per layer was $\lfloor L_a / D_{wire} \rfloor$ and the number of layers was $\left\lfloor \frac{L_b - D_{wire}}{\sqrt{3}D_{wire}/2} \right\rfloor + 1$, so that the accurate value of coil turns was

$$
\begin{aligned}
N &= C'_f \left\lfloor \frac{L_a}{D_{wire}} \right\rfloor \left( \left\lfloor \frac{L_b - D_{wire}}{\sqrt{3}D_{wire}/2} \right\rfloor + 1 \right) \\
&\leq C'_f \left( \frac{L_a L_b}{\sqrt{3}D_{wire}^2/2} - 0.155 \frac{L_a}{D_{wire}} \right) \\
&\approx C_f \frac{L_a L_b}{\pi D_{wire}^2/4}
\end{aligned}
\tag{2}
$$

where $C_f{'}$ was introduced to describing the winding effect, $C_f$ was the filling factor of the enameled wire. $L_a L_b$ was the axis-sectional area of the coil and $\pi D_{wire}^2/4$ was the cross-sectional area of single enameled wire.

From Equation (2), the assumption that $N$ was positively proportional to the ratio of the cross-sectional area of the coil skeleton to $D_{wire}$ was conditional. That was, with the effectiveness of Equation (2), determined by the weight of $C_f'0.155\, L_a/D_{wire}$ in the total coil turns. Additionally, the relative error of the positively proportional function was $0.155/(1.155\, L_b/D_{wire} - 0.155) \times 100\%$, which was determined by $L_b/D_{wire}$.

$L_b/D_{wire}$ determined the number of layers and the relative error, which are displayed in Figure 4. From the calculation results, the relative error of $C_f L_a L_b/(\pi D_{wire}^2/4)$ computing $N$ decreased with $L_b/D_{wire}$ increasing. To guarantee that the relative error of Equation (2) is lower than 5.0% in computing $N$, it should be met that $L_b > 2.8\, D_{wire}$. That is, the coil should be wound with three layers at least. When $L_b < 2.8\, D_{wire}$, one should use $L_b' = \left\lfloor \frac{L_b - D_{wire}}{\sqrt{3}D_{wire}/2} \right\rfloor + 1$ instead of $L_b$ for computations. For the coils in this paper, the values of $L_b/D_{wire}$ under different $D_{wire}$ were higher than 6.8/0.8 = 8.5 so that the approximate expression in Equation (2) has enough precision.

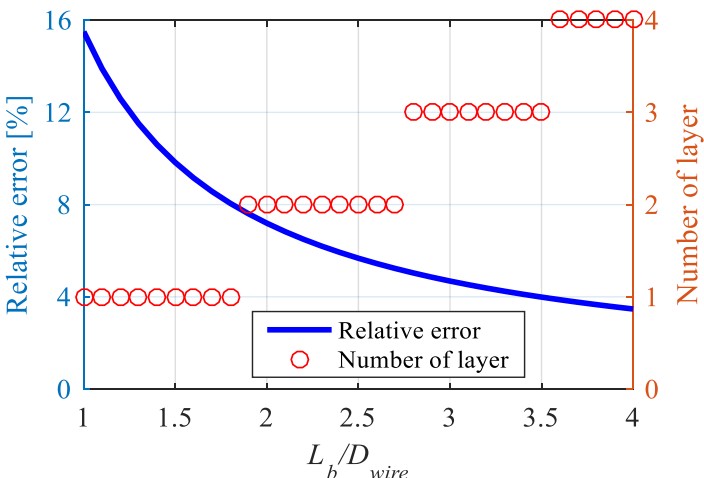

**Figure 4.** Relative errors of the positively proportional function of the available area of coil skeleton vs. the cross-sectional area of enameled wire.

For convenience, the value of $C_f$ was determined by the mean values of $N$ and $D_{wire}$ so that $C_f = 0.57$. The effect of Equation (2) computing $N$ is shown in Table 4. From the calculation results, the model of coil turn can predict the practical coil turn effectively as the relative error was lower than 2.8%.

**Table 4.** Coil turns from the test and model.

| Coil Label | Coil Turns from Test | Coil Turns from Model [1] | Relative Error (%) |
|---|---|---|---|
| 1 | 837 | 847.33 | 1.23 |
| 2 | 537 | 535.36 | −0.30 |
| 3 | 342 | 339.15 | −0.83 |
| 4 | 229 | 226.19 | −1.23 |
| 5 | 175 | 171.03 | −2.27 |
| 6 | 124 | 127.23 | 2.61 |

[1] Cannot be an integer.

From above analysis, the empirical equations in describing the relationships between $D_{core}$, $D_{wire}$ and $N$ were written as

$$\begin{cases} \hat{D}_{core} = 0.962D_{wire} - 0.0277 \text{ or } 0.898D_{wire} \\ N = C_f \dfrac{L_a L_b}{\pi D_{wire}^2/4} \end{cases} \tag{3}$$

### 3.1.2. Static Resistance and Static Inductance

Static resistance and inductance are the key parameters to determine the current response of a coil with an unobvious skin effect. Based on the empirical expression of inductance $L$ and the basic equation of resistance $R$, the model can be easily established as

$$
\begin{cases}
L = 4\pi C_{L0} N^2 = \dfrac{C_L C_f^2}{D_{wire}^4} \\[2mm]
R = \rho \dfrac{N(L_f + L_b)}{D_{core}^2 / 4} = 16\rho C_f \dfrac{L_a L_b (L_f + L_b)}{\pi D_{core}^2 D_{wire}^2}
\end{cases}
\tag{4}
$$

where $C_{L0}$ and $C_L$ were two parameters dependent on $L_a$, $L_b$, $L_f$ while independent of other variables and met $C_L = 64\,C_{L0}(L_a L_b)^2 / \pi$; $\rho$ was the resistivity of copper.

Figure 5 displays the relationships between $L$, $R$, and $D_{wire}$ from the experiment and computation. From the results, it was easily reached that both $L$ and $R$ were monotonically decreasing functions vs. $D_{wire}$. More specifically, as concluded from the expression of $N$ in Equation (3) and $D_{core} = 0.898\,D_{wire}$, both $R$ and $L$ were inversely proportional functions vs. $D_{wire}^4$ (also $N^2$). The model was in good agreement with the experiment as the relative errors of the model in computing $R$ and $L$ were lower than 3.1% and 2.8%, respectively.

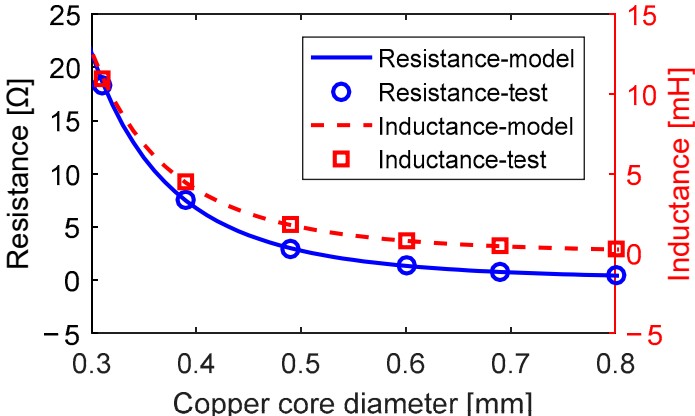

**Figure 5.** Curves of the static resistance and static inductance vs. the enameled wire diameter.

### 3.2. Sinusoidal Response

Equivalent to the series connection of an inductor and a resistor, the electromagnetic coil was generally modeled as a first-order linear time-invariant system model. Additionally, the amplitude-frequency and phase-frequency characteristics are the most important characteristics of the sinusoidal response of the coil.

Stimulated by a sinusoidal voltage $U(t) = U_{amp}\sin(\omega t)$, the current response within the coil can be calculated by $I(t) = I_{amp}\sin(\omega t - \varphi_I)$, where $\omega$ is the angular frequency of the input and $\varphi_I$ is the phase lag of the coil current compared to the voltage. From the theory of the linear time-invariant system, the amplitude ratio function is $A_I = I_{amp}/U_{amp} = 1/(R^2 + \omega^2 L^2)^{1/2}$, and $\tan\varphi_I = \omega L/R$. Substituting Equation (4) into these expressions, one obtains

$$
\begin{cases}
A_I = \dfrac{\pi D_{core}^2 D_{wire}^2}{C_f L_a \sqrt{\left[\dfrac{16\rho L_b \cdot}{(D_f + L_b)}\right]^2 + \left(\dfrac{\omega C_L C_f D_{core}^2}{D_{wire}^2}\right)^2}} \approx \dfrac{0.806\pi D_{wire}^4}{C_f L_a \sqrt{\left[\dfrac{16\rho L_b \cdot}{(D_f + L_b)}\right]^2 + (0.806\omega C_L C_f)^2}} \\[4mm]
\varphi_I = \arctan \dfrac{\omega C_L C_f}{16\rho L_b (D_f + L_b)} \cdot \dfrac{D_{core}^2}{D_{wire}^2} \approx \arctan \dfrac{0.0504\omega C_L C_f}{\rho L_b (D_f + L_b)}
\end{cases}
\tag{5}
$$

From the empirical equation of the mean magnetic field given in Equation (1), the amplitude radio to inputted voltage of the magnetic field $A_H$ and the lagging phase of

the magnetic field $\varphi_H$ can be easily reached as $C_{HI}NA_I/L_a$ and $\varphi_H = \varphi_I$. By substituting Equation (5) into these two equations, one obtains

$$
\begin{cases}
A_H = \dfrac{4C_{HI}L_bD_{core}^2}{L_a\sqrt{\left[\begin{array}{c}16\rho L_b\cdot\\(D_f+L_b)\end{array}\right]^2+\left(\dfrac{\omega C_LC_fD_{core}^2}{D_{wire}^2}\right)^2}} \approx \dfrac{3.226C_{HI}L_bD_{wire}^2}{L_a\sqrt{\left[\begin{array}{c}16\rho L_b\cdot\\(D_f+L_b)\end{array}\right]^2+\left(0.806\omega C_LC_f\right)^2}}\\[4ex]
\varphi_H = \arctan\dfrac{\omega C_LC_f}{16\rho L_b(D_f+L_b)}\cdot\dfrac{D_{core}^2}{D_{wire}^2} \approx \arctan\dfrac{0.0504\omega C_LC_f}{\rho L_b(D_f+L_b)}
\end{cases}
\tag{6}
$$

Changing the frequency from 10 Hz to 1000 Hz, Figure 6 shows the tested and calculated amplitude ratios and phase lags of the magnetic field with respect to the inputted voltage. To demonstrate the influence of the wire diameter more clearly, the wire diameter was plotted on the horizontal axis. From the tested and calculated results, a wider wire diameter is quite helpful for a higher magnetic field amplitude as the amplitude ratio increased faster with an increase in wire diameter. On the contrary, the wire diameter has little influence on the phase lag of the magnetic field, which represents the response time of the magnetic field from 0 to some required proportion of a steady-state value.

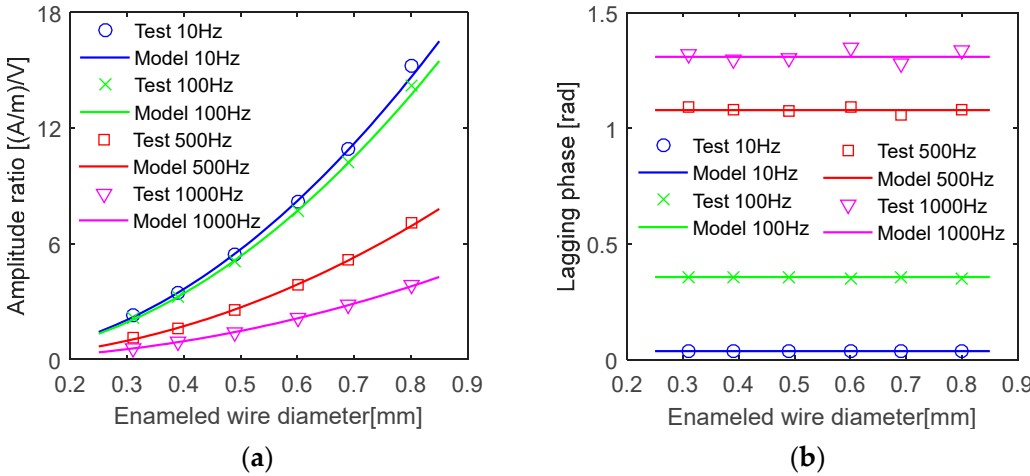

**Figure 6.** Amplitude ratio and phase lag of the magnetic field under different wire diameters: (**a**) curves of the amplitude ratio vs. enameled wire diameter; (**b**) curves of the phase lag vs. enameled wire diameter.

Figure 7 shows the relative errors of the model under various frequencies. For predicting the amplitude ratio, the calculation error was lower than 2.0% when the wire diameter was between 0.39 mm and 0.69 mm. The model accuracy was a little lower when the wire diameter was wider than 0.8 mm or narrower than 0.31 mm, as the relative errors at these points were higher than 5%; this was acceptable as the errors were still lower than 6.4%. For computing the lagging phase, the relative errors under different parameters, including various frequencies and wire diameters, were lower than 3.2%, which showed high precision of the model in predicting the lagging phase of the magnetic field. A low calculation accuracy regarding the computing amplitude ratio was mainly caused by poor winding when the coil wire was quite thin or thick. On the whole, the proposed models for the magnetic field amplitude and lagging phase were verified by the low relative errors under most conditions.

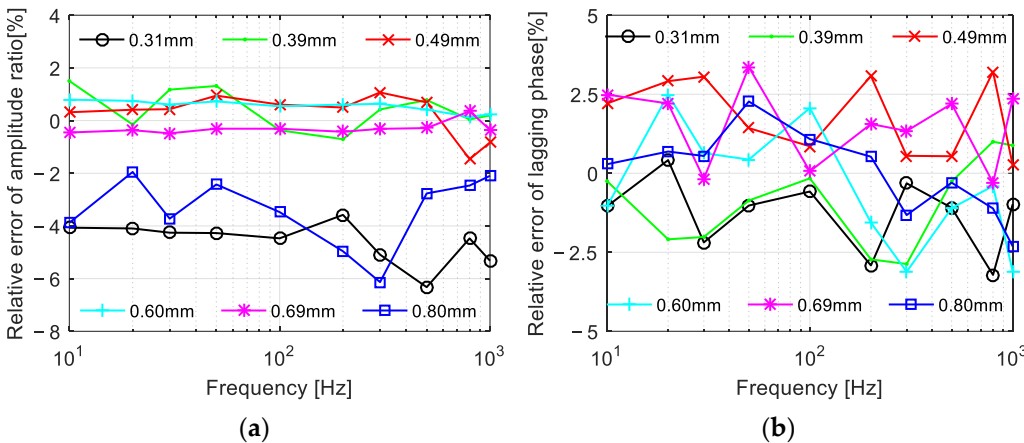

**Figure 7.** The relative errors of the model in computing the coil current under a harmonic voltage: (**a**) the relative errors of computing the amplitude ratio; (**b**) the relative errors of computing the phase lag.

Figure 8 shows the relationships between $A_H$ and $D_{wire}{}^2$. From Figure 8, the linear relationship between the amplitude ratio of the magnetic field and the square of the wire diameter was verified as the tested points under a certain frequency were roughly plotted in a line passing through the origin.

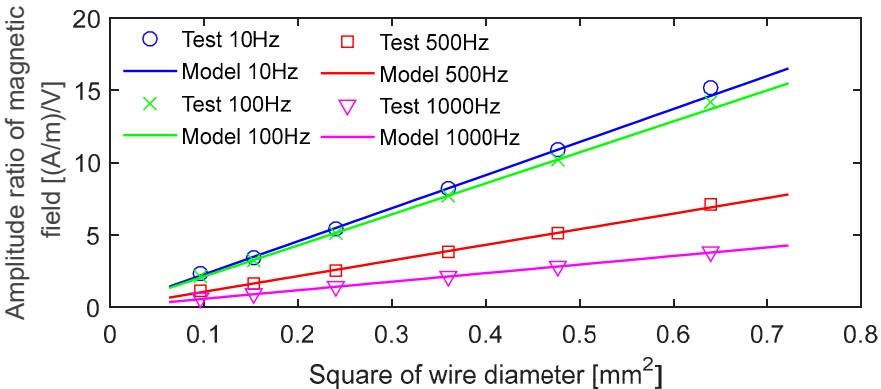

**Figure 8.** Linear curves of $A_H$-$D_{wire}{}^2$ from the model and test under different frequencies.

### 3.3. Square-Wave Response

#### 3.3.1. Time-Domain Response

In addition to the sinusoidal voltage, the direct current (DC) square-wave voltage is frequently used, especially to drive an on–off-type actuator.

For the square-wave response, more attention should be paid to the transient-state process. Additionally, based on the first-order linear time-invariant system, the transient-state current within the coil is

$$I(t) = \frac{U_{st}}{R} + (I_0 - \frac{U_{st}}{R})e^{-\frac{R}{L}t} \tag{7}$$

where $I_0$ is the initial value of the coil current, $U_{st}$ is the steady-state amplitude of the voltage. Equation (7) was suitable to both the charging and discharging process of the coil. For charging, $I_0 = 0$. For discharging, $U_{st} = 0$.

From Equation (4), the reciprocal of the time-constant used in Equation (7) was

$$\frac{R}{L} = \frac{\rho(L_f + L_b)D_{wire}^2}{4C_{L0}C_f L_a L_b D_{core}^2} \tag{8}$$

By substituting Equations (7) and (8) into $H(t) = C_{HI}NI(t)/L_a$, one obtains the transient-state response of the magnetic field

$$H(t) = \frac{C_{HI}C_f L_b}{\pi D_{wire}^2/4}\left[\frac{U_{st}}{R} + (I_0 - \frac{U_{st}}{R})e^{-\frac{R}{L}t}\right]$$

$$= \frac{4C_{HI}C_f L_b}{\pi D_{wire}^2}\left[\frac{\pi D_{core}^2 D_{wire}^2 U_{st}}{16\rho C_f L_a L_b(L_f + L_b)} + \left(I_0 - \frac{\pi D_{core}^2 D_{wire}^2 U_{st}}{16\rho C_f L_a L_b(L_f + L_b)}\right)e^{-\frac{\rho(L_f+L_b)D_{wire}^2}{4C_{L0}C_f L_a L_b D_{core}^2}t}\right] \quad (9)$$

$$= \frac{C_{HI}D_{core}^2}{4\rho L_a(L_f + L_b)}U_{st} + \frac{4C_{HI}C_f L_b}{\pi D_{wire}^2}I_0 e^{-\frac{\rho(L_f+L_b)D_{wire}^2}{4C_{L0}C_f L_a L_b D_{core}^2}t} - \frac{C_{HI}D_{core}^2}{4\rho L_a(L_f + L_b)}U_{st}e^{-\frac{\rho(L_f+L_b)D_{wire}^2}{4C_{L0}C_f L_a L_b D_{core}^2}t}$$

The inputted voltage was generated with an amplitude of 2 V and a high-voltage duration of 20 ms to guarantee the coil current reaching the steady state. The time-domain magnetic fields are shown in Figure 9. From the test and model, the proposed model precisely calculated the amplitudes and effectively described the curve shapes under different wire diameters as the transient-state results were also quite close.

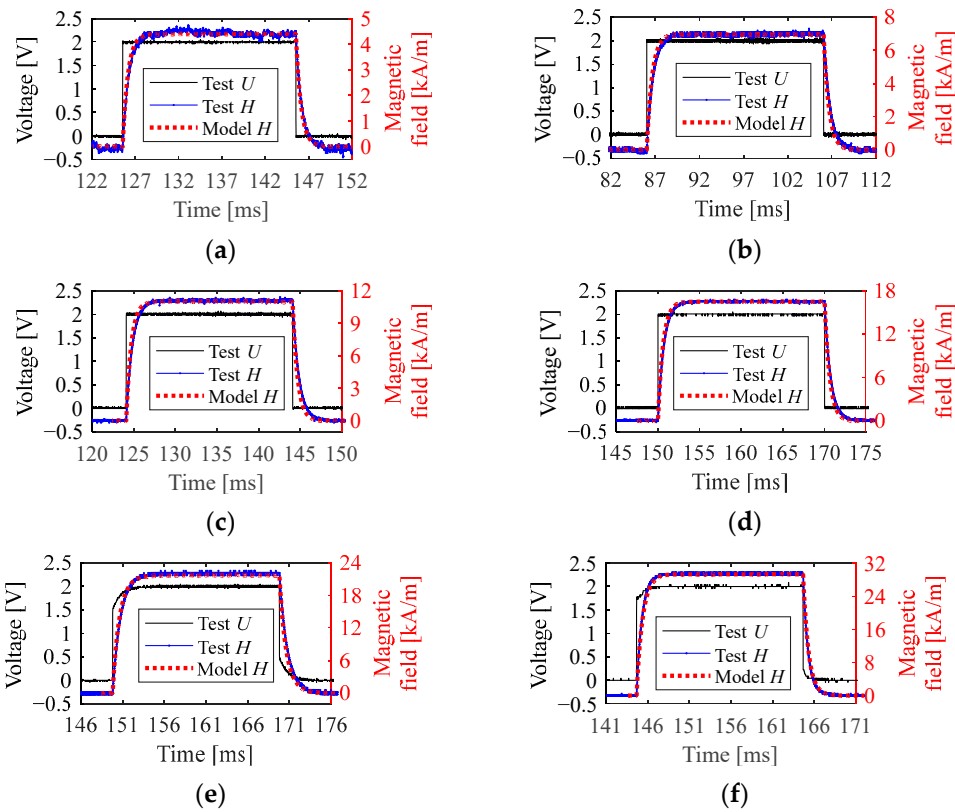

**Figure 9.** The dynamic magnetic field under square-wave input: (**a**) $D_{wire} = 0.31$ mm; (**b**) $D_{wire} = 0.39$ mm; (**c**) $D_{wire} = 0.49$ mm; (**d**) $D_{wire} = 0.60$ mm; (**e**) $D_{wire} = 0.69$ mm; (**f**) $D_{wire} = 0.80$ mm.

3.3.2. Steady-State Value and Response Time

The steady-state value and response time are the most important characteristic parameters of the on–off-type actuator. When the square-wave voltage maintains a high level for a long enough duration, the steady-state response of the magnetic field $H_{st}$ can be easily acquired from Equation (9), as

$$H_{st} = \frac{C_{HI}D_{core}^2}{4\rho L_a(L_f + L_b)}U_{st} \approx \frac{0.2016 C_{HI}D_{wire}^2}{\rho L_a(L_f + L_b)}U_{st} \quad (10)$$

By exacting the mean value of the magnetic field in the steady-state stage, Figure 10 shows the curves of $H_{st}$ vs. $D_{wire}$ from the test and model. From the tested and calculated results, a higher $D_{wire}$ is helpful for a higher $H_{st}$. More specifically, $H_{st}$ was positively proportional to $D_{wire}{}^2$, as explained in Equation (10). Thus, the change law of the $H_{st}$ under the square wave was the same as one of the magnetic field amplitudes under the sinusoidal voltage. It was easily illustrated that both the functions of $1/(R^2 + \omega^2 L^2)^{1/2}$ and $1/R$ can be expressed by the quartic function vs. the wire diameter approximately. In addition, the relatively errors under different wire diameters were less than 1.2% thus, the model can predict the steady-state magnetic field quite effectively.

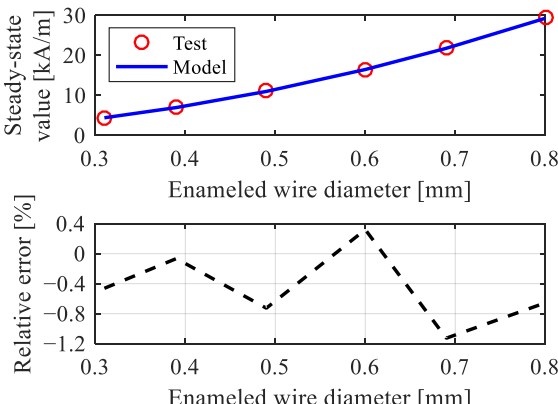

**Figure 10.** Steady-state magnetic field of the square-wave response from the test and model.

There are two commonly used response times—the time from 0 to the required intensity and the time from 0 to the required proportion of the steady-state value. The former was especially concerned when the high-opening voltage was employed and the latter was generally concerned when the standard square-wave voltage was employed (or the duty cycle was adjusted but not the amplitude in the voltage wave).

By imposing $I_{req}$ the required intensity of the coil current and substituting $I_{req}$ into Equation (7), the response time $t_{Iv}$ can be reached, as

$$
\begin{aligned}
t_{Iv} &= \frac{L}{R} \ln\left(1 + \frac{I_{req}}{U_{st}/R - I_{req}}\right) \\
&= \frac{C_L C_f}{16\rho L_b (D_f + L_b)} \frac{D_{core}^2}{D_{wire}^2} \ln\left(1 + \frac{I_{req}}{\pi D_{core}^2 D_{wire}^2 U_{st}/\left[16\rho C_f L_a L_b (D_f + L_b)\right] - I_{req}}\right)
\end{aligned}
\tag{11}
$$

Similarly, by substituting the required magnetic field $H_{req}$ into Equation (9), one obtains the response time to the specified magnetic field intensity $t_{Hv}$, as

$$
\begin{aligned}
t_{Hv} &= \frac{L}{R} \ln\left(1 + \frac{L_a H_{req}}{C_{HI} N U_{st}/R - L_a H_{req}}\right) \\
&= \frac{C_L C_f}{16\rho L_b (D_f + L_b)} \frac{D_{core}^2}{D_{wire}^2} \ln\left(1 + \frac{L_a H_{req}}{C_{HI} D_{core}^2 U_{st}/\left[4\rho(D_f + L_b)\right] - L_a H_{req}}\right) \\
&\approx \frac{0.0504\omega C_L C_f}{\rho L_b (D_f + L_b)} \ln\left(1 + \frac{L_a H_{req}}{0.2016 C_{HI} D_{wire}^2 U_{st}/\left[\rho(D_f + L_b)\right] - L_a H_{req}}\right)
\end{aligned}
\tag{12}
$$

From the calculated result, it can be observed that a thicker wire is better for reducing both the response time of the coil current and one of the magnetic fields, while the change degree is different. The enameled wire diameter has more influence on the response time of the coil current than that of the magnetic field as $t_{Iv}$ is in the function form of $a\ln[1 + b/(cx^2 - b)]$ vs. $D_{wire}$ while $t_{Hv}$ is expressed by $a\ln[1 + b/(cx^4 - b)]$ vs. $D_{wire}$.

The response time to a specified proportion of the steady-state value can be easily deduced from Equations (10) and (12). For a given proportion $p$, the corresponding intensities of the coil current and magnetic field are, respectively, $I_{req} = p(U_{st}/R)$ or $H_{req} = p[C_{HI}N(U_{st}/R)/L_a]$. By substituting the two expressions into Equations (10) and (11), one obtains the response time to a specified proportion, as

$$
\begin{aligned}
t_{Hp} = t_{Ip} &= \frac{L}{R} \ln\left(\frac{1}{1-p}\right) \\
&= \frac{C_L C_f}{16\rho L_b (D_f + L_b)} \frac{D_{core}^2}{D_{wire}^2} \ln\left(\frac{1}{1-p}\right) \\
&\approx \frac{0.0504\omega C_L C_f}{\rho L_b (D_f + L_b)} \ln\left(\frac{1}{1-p}\right)
\end{aligned}
\tag{13}
$$

where $p$ is a constant belonging to (0, 1).

Compared to $t_{Hv}$, the factors influencing $t_{Hp}$ were almost independent of $D_{wire}$. More specifically, $t_{Hp}$ was just determined by the ratio of $L/R$. The value of $L/R$ was only slightly influenced by $D_{wire}$; optimizing the wire diameter would be helpless to promote this type of response speed.

Figure 11 shows the two types of response times from the tested and calculated results; the specified intensities $H_{req}$ were 3 kA/m, 3.5 kA/m, and 4 kA/m, and the specified proportions $p$ were 0.7, 0.8, and 0.9. Just as predicted by the model, $H_{req}$ was effectively reduced by increasing $D_{wire}$. Furthermore, $H_{req}$ declined fast first and then slowly with $D_{wire}$ increasing. For the value of $t_{Hp}$, it changed slightly with $D_{wire}$ increasing. The model was verified as the calculated results were consistent with the experimental data.

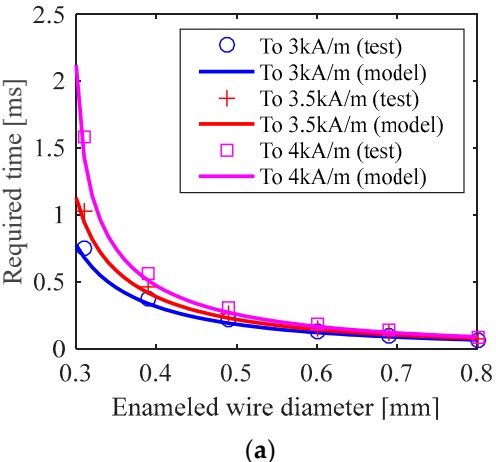
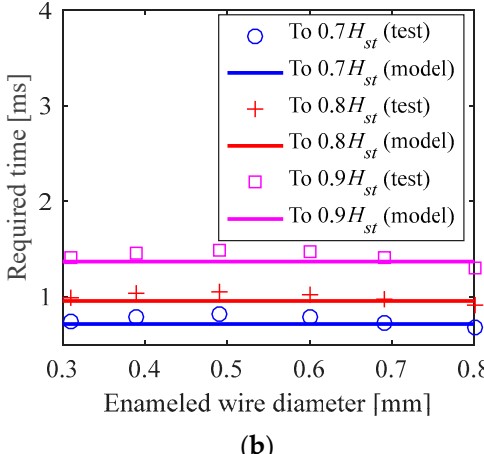

(**a**)               (**b**)

**Figure 11.** The response time of the square-wave response from the test and model: (**a**) the time from 0 to specified intensities, respectively, of 3 kA/m, 3.5 kA/m, and 4 kA/m; (**b**) the time from 0 to the specified proportions of the steady-state response, respectively, of 0.7, 0.8, and 0.9.

On the whole, increasing the wire diameter is quite helpful for reducing the response time from 0 to a specified value of the coil current or magnetic field, while failing to improve the response speed from 0 to the steady-state or any other proportional value. Therefore, for an electromagnetic actuator stimulated by a high-open-low-hold-type voltage, a coil with a wider wire diameter will be stimulated more quickly to save the response time of the whole actuator, while when a traditional square-wave voltage is introduced, an adjustment in the wire diameter is helpless.

## 4. Conclusions

An analytical model of the mean magnetic field for the hollow cylindrical coil used in a magnetically driven actuator was proposed in this paper. Additionally, the selection of the

enameled wire diameter was optimized for a high-amplitude and fast-response magnetic field based on the model.

(1) The resistance and inductance are inversely proportional functions vs. the quartic of the enameled wire diameter. Under the sinusoidal voltage, a wider wire diameter is quite helpful for a higher magnetic field amplitude while it has little influence on the phase lag of the magnetic field. Under the square-wave voltage, the steady-state magnetic field was positively proportional to the square of the wire diameter, as a wider wire diameter is helpful for a higher steady-state magnetic field. Regarding the response speed, increasing the wire's diameter is helpful for reducing the response time from 0 to the specified intensity, while it is helpless to improve the response speed from 0 to the steady-state or any other proportional value.

(2) The proposed model was verified as the calculated results from the model were in good agreement with the experimental results. Specifically, the relative errors of the model in computing the resistance and the inductance were lower than 3.1% and 2.8%, respectively. For predicting the sinusoidal response, the errors were lower than 6.4% (lower than 2.0% under most conditions) in computing the amplitude and lower than 3.2% in computing the lagging phase. For predicting the square-wave response, the model calculated the amplitudes with errors lower than 1.2% and described the curve shape effectively.

This paper was devoted to the promotion of the output performance of the whole magnetically driven actuator without considering the coil quality factor or power loss. Further work can focus on reducing the power loss of the coil.

**Author Contributions:** Conceptualization. G.X.; Data curation, Z.W.; Formal analysis, Z.W.; Investigation: Z.W.; Software and visualization, Z.W. and G.X.; Methodology, G.X.; Validation, G.X. and H.B.; Funding acquisition, H.B. and G.X.; Writing—review and editing, Z.W. and G.X.; Writing—review and editing, H.B., G.X. and Z.R.; Supervision: Z.R. All authors have read and agreed to the published version of the manuscript.

**Funding:** This work was supported by the Young and Middle-aged Teachers Education and Research Project (Science and Technology) of Fujian Province (No. JAT220016) and First Batch of Yin Ling Fund (No. ZL3H39).

**Data Availability Statement:** The data presented in this study are available on request from the corresponding author.

**Acknowledgments:** The authors thank Zhaoshu Yang (in China Astronaut Research and Training Center) and Tuo Li (in Officers College of PAP) for their improvement in the English of this article.

**Conflicts of Interest:** The authors declare no conflict of interest.

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
