# Peer review of "Optimization of the Wire Diameter Based on the Analytical Model of the Mean Magnetic Field for a Magnetically Driven Actuator"

_aerospace, doi:10.3390/aerospace10030270_

Round 1

Reviewer 1 Report

This paper presents an optimization method to improve dynamic performance of electromagnetic solenoid actuator for aerospace applications. The topic is interesting but there are several weaknesses that need to be addressed:

- paper needs to clearly define what aspects of the actuator design are they trying to optimize.  The introduction does not make this clear. Is this discussion for solenoid actuators ?  This needs to be made clear.

- paper discusses only air core solenoid actuators. This is a major limiting factor . Most practical applications require iron core solenoids, and this case the analysis provided would not be useful due to nonlinearities created by iron cores.

- line 106: should say "induced" magnetic field

- line 112: what are the units of CHI ? equation 1 needs to be explained or provide a reference

- Figure 1 (a) has a significant mistake.  An oscilloscope only READS signals, it cannot generate a signal to drive a power amplifier. 

- Is the power amplifier in Fig 1(a) a voltage amplifier or current amplifier ?

- is the measurement on coil voltage done differential or single ended ?

- the title of the paper implies analytical optimization. An optimization implies a clear set of performance metric.  This has not been defined in the paper . What is being optimized ?

- optimization also implies a mathematical procedure to optimize a performance metric relative to a variable ( in this case, wire diameter).  This is not presented.

In summary, this paper addresses an important topic but the method is not adequate and the results are therefore not clear.  

Reviewer 2 Report

First of all, thank you very much for the publication, for the material you sent.

This is a very interesting and desirable publication.

The research itself and the methodology are correct. So is the mathematical apparatus.

Nevertheless, please highlight why aerospace is the authors' target field.

If this is the use, please highlight the problems related to the work of coils, electromagnets in the air and outer space. Not only electrical but also mechanical research. Increasing the degree of reliability increases the number of electrical and mechanical tests.

Best regards.

Reviewer 3 Report

The paper presents a study on the effects of wire diameter on the response of a solenoid. I think that a major revision is needed before the manuscript can be considered for publication. I suggest that the authors address the points listed below:

1. The abstract and introduction should contextualize the subject of the study within the scope of the journal, for example with a brief review of the potential application of solenoid actuators in the aerospace field.

2. It is not clear what is the original contribution of the paper. In the literature review, the authors shall highlight the novelty of their work and how it contributes to fill a gap in the current state-of-the-art design process for solenoid actuators.

3. Figure 5: I suggest having iso-wire-diameter curves of amplitude ratio vs frequency and phase vs frequency (so basically a Bode diagram foe each wire diameter) - like what is already done for Figure 6. In my opinion, this would improve the readabilit of the figure.

4. Figure 8: why the measured input voltage "Test U" is a square wave for all figures but (e) and (f)?

Round 2

Reviewer 1 Report

The authors have addressed most of the previous comments. There are still a few important issues that need addressing:

Figure 1 is not clear and needs to be improved. The third block (efficient utilization and coil optimization)  does not make sense and needs more clear explanation.  What is "efficient utilization" ?   How is the magnetic circuit optimized ?  How is the coil optimized ?

The main pending issue in this paper is to provide a clear description of what is optimized, what are the variables being optimized and how is the optimization being done.  This should be clearly stated before Section 2.

Reviewer 3 Report

The authors addressed all the comments from my previous review, so I suggest that the paper can be accepted in present form.

Author Response

Thanks for your comments. The comments are of great help to improve our paper.

Round 3

Reviewer 1 Report

The authors have addressed my concerns and comments. However, there are some important figures, diagrams and explanations that were included in the letter to the reviewer but have not been included in the paper.  These should be added to the paper, as they provide important explanation of the actuator being used and the optimization process.
